# Effects of Dietary Carbohydrate Concentration and Glycemic Index on Blood Glucose Variability and Free Fatty Acids in Individuals with Type 1 Diabetes

**DOI:** 10.3390/nu16091383

**Published:** 2024-05-02

**Authors:** Selda Seckiner, Murat Bas, Ilgin Yildirim Simsir, Su Ozgur, Yasemin Akcay, Cigdem Gozde Aslan, Ozge Kucukerdonmez, Sevki Cetinkalp

**Affiliations:** 1Department of Nutrition and Dietetics, Faculty of Health Sciences, İstanbul Beykent University, Istanbul 34520, Turkey; 2Institute of Health Sciences, Acibadem Mehmet Ali Aydınlar University, Istanbul 34450, Turkey; 3Department of Nutrition and Dietetics, Faculty of Health Sciences, Acibadem Mehmet Ali Aydınlar University, Istanbul 34450, Turkey; murat.bas@acibadem.edu.tr; 4Division of Endocrinology and Metabolism Disorders, Department of Internal Medicine, Faculty of Medicine, Ege University, Izmir 35100, Turkey; ilginyildirim@hotmail.com (I.Y.S.); scetinkalp@hotmail.com (S.C.); 5Department of Biostatistics and Medical Informatics, Faculty of Medicine, Ege University, Izmir 35100, Turkey; suozgur35@gmail.com; 6Department of Medical Biochemistry, Faculty of Medicine, Ege University, Izmir 35100, Turkey; yasemin.akcay@ege.edu.tr; 7Department of Medical Biochemistry, Faculty of Medicine, Biruni University, Istanbul 34010, Turkey; cigdemgozdeaslan@gmail.com; 8Department of Nutrition and Dietetics, Faculty of Health Sciences, Ege University, Izmir 35100, Turkey; dytozgek@hotmail.com

**Keywords:** carbohydrate, free fatty acid, glycemic index, glycemic load, glycemic variability, type 1 diabetes

## Abstract

Monitoring glycemic control status is the cornerstone of diabetes management. This study aimed to reveal whether moderate-carbohydrate (CHO) diets increase the risk of free fatty acid (FFA) levels, and it presents the short-term effects of four different diet models on blood sugar, glycemic variability (GV), and FFA levels. This crossover study included 17 patients with type 1 diabetes mellitus to identify the effects of four diets with different CHO contents and glycemic index (GI) on GV and plasma FFA levels. Diet 1 (D1) contained 40% CHO with a low GI, diet 2 (D2) contained 40% CHO with a high GI, diet 3 (D3) contained 60% CHO with a low GI, and diet 4 (D4) contained 60% CHO with a high GI. Interventions were performed with sensor monitoring in four-day periods and completed in four weeks. No statistical difference was observed among the groups in terms of blood glucose area under the curve (*p* = 0.78), mean blood glucose levels (*p* = 0.28), GV (*p* = 0.59), and time in range (*p* = 0.567). FFA and total triglyceride levels were higher in the D1 group (*p* < 0.014 and *p* = 0.002, respectively). Different diets may increase the risk of cardiovascular diseases by affecting GI, FFA, and blood glucose levels.

## 1. Introduction

Monitoring the glycemic control status is the cornerstone of diabetes management. Glucose data analysis enables the assessment of treatment efficacy and guides lifestyle and medication adjustments to safely achieve the best possible blood sugar control [1].

The American Diabetes Association dietary guidelines state that there is no “ideal” macronutrient ratio of carbohydrates (CHOs), proteins, and fats for individuals with diabetes, whereas the International Society for Pediatric and Adolescent Diabetes suggests that 40–50% of energy needs should be met with CHOs, <40% with fats (<10% saturated fat + trans fatty acids), and 15–25% with proteins. However, changes should be made according to individual characteristics. The number of studies showing that low-CHO diets, which are not evidence-based but frequently used in nutrition therapy in type 1 diabetes mellitus (T1DM), improve glycemic outcomes and body weight control is limited [2,3]. However, low-CHO diets are regaining popularity worldwide; the common definition is that <26% of an individual’s total daily energy intake is from CHO. In adults with T1DM, low-CHO diets reduced postprandial hyperglycemia, improved glycated hemoglobin (HbA1c), increased the time in interval, and reduced the risk of hypoglycemia due to reduced insulin requirements [4]. Studies wherein meal skipping was avoided and a balanced macronutrient intake was realized reported that better metabolic control was achieved and glycemic control was improved [5,6]; therefore, individuals with diabetes should consume foods that set a slower and more stable glycemic peak [7]. Moreover, the recommendation for individuals using continuous glucose monitoring is to spend >70% of the time with sensor glucose values in the range (TIR) (3.9–10.0 mmol/L), <25% of the time above the range (TAR) (>10.0 mmol/L), and <4% of the time below the range (TBR) (<3.9 mmol/L) [8]. Furthermore, the glycemic variability (GV) expressed as the coefficient of variation should not exceed 36%.

Plasma free fatty acid (FFA) levels increase with low CHO intake and decrease with high CHO intake. Most studies on this topic reported that diets with high CHO intake and low GI positively affected insulin sensitivity. Increasing FFA levels inhibit peripheral glucose uptake and increase ketone production, whereas increased ketone production causes ketoacidosis if its production exceeds the peripheral utilization and renal excretion capacities [6].

In medical nutrition therapy, the content of CHOs, as well as the meal composition, can reduce metabolic control and the risk of complications. Diets with moderate CHO intake and low GI are essential for blood glucose variability and cardiovascular disease (CVD) risk in individuals with T1DM.

This study aimed to reveal the short-term effects of four different diet models (D1, 40% CHO + low GI; D2, 40% CHO + high GI; D3, 60% CHO + low GI; and D4, 60% CHO + high GI) on blood glucose levels, GV (also known as %CV), and FFA levels.

## 2. Materials and Methods

This study was conducted with individuals who were followed up with for T1DM at the Endocrinology Outpatient Clinic between September 2020 and March 2021. This study was planned as a crossover study with a total of 20 cases of T1DM with a power of 80% and an error margin of 0.05.

Patients who were on multiple-dose insulin therapy (detemir, glargine U-100, or glargine U-300 as basal insulin; lispro, aspart, or glulisine as bolus); receiving a daily insulin dose of ≥0.5 IU/kg/day (those not in the honeymoon phase); and had a diabetes duration >1 year, an HbA1c level of ≤10%, and a body mass index (BMI) of ≤30 kg/m^2^ were included in this study. Patients who had a disease requiring food restriction (diabetic nephropathy, celiac, food allergy, and eating behavior disorder), had chronic complications of diabetes, used additional drugs/vitamins other than insulin, received neutral protamine Hagedorn as basal insulin or regular insulin as bolus, received two detemir or glargine doses daily, used three doses of insulin or less daily, were believed to be in the honeymoon phase (insulin requirement <0.5 IU/kg/day), had an HbA1c level of >10%, had a BMI of >30 kg/m^2^, did not accept to participate voluntarily, or were pregnant were excluded.

This study, which started with 20 participants, was completed with 17 participants because 1 person had a measurement error, and 2 people voluntarily withdrew participation.

This study was approved in terms of medical ethics with decision of the Ethics Committee of Acibadem University, (ATADEK) with the decision number 2019-12/24, dated 11 July 2019. Each participant read and signed an informed consent form.

To increase the reliability of our study, blood glucose monitoring was performed using the closed continuous glucose measurement system (CGMS) using the Medtronic (Macquarie Park, NSW, Australia) iPro ^TM^ 2 ABDsensor. To ensure standardization in the diet menus, a special food preparation and distribution system was used. All the foods on their menu were provided. During the four diet models provided to individuals with diabetes, energy intake, number of meals of the diet, and mealtimes of the diet were prepared similarly to each other. The glycemic index (GI) was considered primarily in the content of the diets, and menu planning was not performed on the basis of the glycemic load (GL).

All individuals with T1DM were provided with individually calculated diets with a standard CHO amount and content. Each dietary set was completed in 4 weeks, with 5 days of administration and 2 days of washout. Diet 1 (D1) contained 40% CHO with a low GI, diet 2 (D2) contained 40% CHO with a high GI, diet 3 (D3) contained 60% CHO with a low GI, and diet 4 (D4) contained 60% CHO with a high GI. The sample diet contents are presented in Appendix A; their macronutrient contents and percentages are presented in Appendix A and Appendix A, respectively; and the research flow chart is presented in Appendix A. All dietary practices were sequentially applied. Continuous glucose monitors (CGMs) were kept on the patients throughout the diet programs. A 2-day washout period was allowed between each diet model. Dietary compliance with scale measurements was tracked via WhatsApp version 2.24.6.78. GI and GL values of all meals are shown in Appendix A.

Each set of sensors was applied on the first day at the beginning of the week, and their biochemical parameters were examined at the end of the 4 days. Regarding biochemical parameters (FFAs, triglycerides [TGs], ketones, and fructosamines), blood was drawn before the intervention and after each dietary treatment. Only ketones were measured after D1 and D2, which contain 40% CHO.

To ensure standardization during the study, it was recommended not to exercise in addition to activities of daily living, and alcohol consumption was not allowed during the study. Individuals were reminded of this during the controls. At the beginning of D1 and D3 diet cycles, body weights were measured using a standard scale.

The Harris–Benedict formula was used for energy calculation; however, ±100 kcal/day change was allowed for energy intake, considering the nutritional habits of the individuals. Although weight loss was not directly sought, moderate calorie restriction was pursued in terms of weight loss for individuals with overweight.

The mean and standard deviation (SD) values of the sample diet menus provided to individuals in terms of energy and macronutrients are listed in Appendix A.

### Statistical Analyses

Statistical analyses were performed using IBM Statistical Package for the Social Sciences v.25 (IBM Corp., Armonk, NY, USA). Descriptive characteristics were presented as numbers, percentages, means ± SDs, and medians (minimum–maximum) values. Before the statistical analyses, the normality of the continuous variables was checked. Analyses were performed using parametric tests in cases wherein data were normally distributed and with nonparametric tests in other situations. Pearson’s chi-square test/Fisher’s exact test were used for evaluating categorical data, whereas McNemar’s test was used for comparing dependent categorical data. In the evaluation of repeated measurements, the Bonferroni test was used for post hoc comparisons in cases wherein statistical significance was determined by applying the repeated measures ANOVA test. Statistical evaluations were performed at the *p* < 0.05 significance level.

## 3. Results

### 3.1. Baseline Characteristics

Of the 17 participants who completed the study, 9 (52.9%) were males and 8 (47.1%) were females, with a mean age of 29.7 ± 10.0 years. At the beginning of the study, the participants had a mean HbA1c level of 7% ± 0.9%, mean systolic blood pressure of 114.7 ± 11.5 mmHg, and diastolic blood pressure of 72.4 ± 8.0 mmHg. Four (23.5%) and nine (52.9%) participants were smokers and alcohol drinkers, respectively. The participants had a mean body weight, BMI, fat mass, fat percentage, and lean body mass of 72.5 ± 17.6 kg, 24.2 ± 3.8 kg/m^2^, 15.5 ± 8.5 kg, 21.2 ± 9.4, and 57.0 ± 15.4 kg, respectively. The sociodemographic data of the participants are presented in Table 1. The baseline body weights were 79.4 ± 15.1 and 62.7 ± 17.0 kg in male and female participants, respectively. Body weights at the end of D1 + D2 were 79.4 ± 14.9 and 61.8 ± 16.6 kg in male and female participants, respectively, and 78.6 ± 15.0 and 61.3 ± 16.5 kg in male and female participants, respectively, at the end of D3 + D4. The difference in body weights among the groups was statistically significant (*p* = 0.03) (Table 2).

The baseline body fat percentages were 16.6 ± 6.2% and 27.7 ± 9.8% in male and female participants, respectively. The fat percentage values at the end of D1 + D2 were 16.8 ± 7.1% and 27.1 ± 9.8% in males and females, respectively, and 16.5 ± 6.4% and 26.6 ± 10.0% in males and females at the end of D3 + D4, respectively. No statistically significant difference was noted between the groups (Table 2).

Regarding the baseline body fat mass, males had a mean mass of 13.3 ± 5.8 kg and females 18.7 ± 11.1. Fat mass values at the end of D1 + D2 were 13.6 ± 6.9 and 18.1 ± 11.1 kg in males and females, respectively, and 13.2 ± 6.2 and 17.6 ± 11.1 kg in males and females at the end of D3 + D4, respectively. No statistically significant difference was observed between the groups (Table 2).

The baseline lean body mass was 66.1 ± 13.0 and 44.0 ± 6.3 kg in male and female participants, respectively. Body lean mass values at the end of D1 + D2 were 65.8 ± 12.5 and 43.7 ± 6.2 kg in males and females, respectively, and 65.5 ± 12.7 and 43.7 ± 6.2 kg in males and females at the end of D3 + D4, respectively. Only the gender parameter showed a statistically significant difference between the groups (*p* = 0.001) (Table 2).

### 3.2. Intended D1 Results

Based on the data obtained using CGMSs, mean blood glucose levels were 135.2 ± 23.0, 138.2 ± 31.1, 141.4 ± 26.5, and 146.8 ± 30.3 mg/dL for patients administered with D1, D2, D3, and D4, respectively. No statistically significant difference in mean blood glucose levels was observed between the groups (*p* = 0.28) (Table 3).

The GV was 34.2 ± 9.8%, 34.4 ± 8.8%, 36.3 ± 9.5%, and 36.8 ± 8.2% after D1, D2, D3, and D4, respectively. No statistically significant difference in GV after dietary interventions was noted (*p* = 0.59) (Table 3).

TIR (Dec value; 70–180 mg/dL) results were 71.7 ± 14.2%, 70.2 ± 16.8%, 69.8 ± 15.4%, and 66.8 ± 14.5% after D1, D2, D3, and D4, respectively. No statistically significant difference was noted between the dietary groups in terms of TIR (*p* = 0.57) (Table 3).

TBR (Dec value; <70 mg/dL) results were 5.8 ± 6.5%, 4.7 ± 3.7%, 5.7 ± 3.8%, and 5.3 ± 3.8% after D1, D2, D3, and D4, respectively. No statistically significant difference was observed between the dietary groups in terms of TBR (*p* = 0.92) (Table 3).

TAR (Dec value; >180 mg/dL) results were 10.2 ± 6.5%, 11.2 ± 9.2%, 11.9 ± 8.0%, and 13.5 ± 7.9% after D1, D2, D3, and D4, respectively. The groups showed no statistically significant difference after dietary interventions (*p* = 0.33) (Table 3).

Since fructosamine levels reflect the average of 2–3 weeks of glycemia, it was evaluated twice at the end of D1 + D2 and D3 + D4. Although the participants’ initial fructosamine levels were 0.4 ± 0.1 µmol/L, they were 0.3 ± 0.0 and 0.3 ± 0.0 µmol/L after D1 + D2 and D3 + D4, respectively. Compared with baseline, no statistically significant difference after dietary interventions was observed (*p* = 0.11) (Table 3).

### 3.3. Intended D2 Results

The mean TG levels of the patients were 73.6 ± 31.3 mg/dL at baseline and 134.3 ± 91.9, 109.4 ± 65.7, 118.8 ± 53.5, and 97.5 ± 61.0 mg/dL after D1, D2, D3, and D4, respectively, exhibiting a statistically significant difference between the groups (*p* = 0.002). In the pairwise comparison of the groups, the mean TG value in D1 was significantly higher than that of D2 (*p* = 0.049). The mean TG level in D1 was significantly higher than those of the other groups (*p* = 0.002) (Table 3).

The patients’ FFA levels were 1.19 ± 0.21 nmol at baseline and 1.60 ± 0.57, 1.36 ± 0.53, 1.41 ± 0.42, and 1.23 ± 0.53 nmol after D1, D2, D3, and D4, respectively. A statistically significant difference was noted between the groups in terms of FFA levels (*p* = 0.018). The significance level between D1 and D4 in the pairwise comparison was *p* = 0.014 (Table 3).

When the intervention groups containing different CHO and GI levels were compared with the blood glucose area under the curve (AUC) averages of the individuals, no statistically significant difference was noted between the groups (*p* = 0.78) (Table 3). Ketone levels were only measured in diets that had low CHO contents, that is, containing 40% CHOs. No statistically significant difference was observed between ketone levels at baseline and after D1 and D2 interventions (*p* = 0.22 and *p* = 0.38, respectively) (Table 3).

The mean blood sugar levels of all groups in D1–D2, D3–D4, and D1–D3 comparisons and the mean and median values of GV, TIR, TBR, TAR, TG, and FFA of all dietary interventions are presented in Table 4. D1D2_average glucose (AG) was 136.68 ± 25.71 mg/dL, D3D4_AG 144.12 ± 25.19 mg/dL, D1D3_AG 138.29 ± 22.62 mg/dL, D2D4_AG 142.5 ± 26.7 mg/dL, D1D2_GV 34.27 ± 8.00%, D3D4_GV 36.52 ± 7.78%, D1D3_GV 35.22 ± 8.11%, D2D4_GV 35.56 ± 7.31%, D1D2_TIR 70.97 ± 13:29%, D3D4_TIR 68.3 ± 12:56%, D1D3_TIR 70.78 ± 13.0%, and D2D4_TIR 68.48 ± 13.02%. No significant difference was detected between the groups. Inter-diet TG measurements were D1D2_TG 121.82 ± 75.76%, D3D4_TG 108.12 ± 53.51%, D1D3_TG 126.53 ± 70.9%, and D2D4_TG 103.41 ± 61.44%. An evaluation of GI alone showed that the high-GI diet groups had lower TG values than the low-GI groups (*p* = 0.022) (Table 4).

Statistically significant differences in insulin doses were observed when comparing low-GI diet models to high-GI models. When the amount of CHO in the meals increased, the insulin doses did not change. However, the insulin doses increased in the diet menus with a high GI (Table 5). The relationship between the diet groups and the total insulin doses administered is depicted in Figure 1.

## 4. Discussion

In the current study, four different dietary interventions were evaluated in terms of GV, insulin doses, FFA levels, TG, ketone, and fructosamine levels. Moderate CHO with LGI is important for GV. Mealtime insulin doses were lower in LGI diets.

The diet plan for individuals with diabetes should focus on macronutrients that have a direct impact on blood glucose levels, lipid profile, and body weight. However, no single ideal dietary energy distribution is noted between CHOs, fats, and proteins for individuals with diabetes. Therefore, an individualized nutrition plan should be developed by considering total energy and metabolic targets [2,9,10].

Studies reported that lowering the amount of CHO is more effective in fluctuations in blood sugar in individuals with T1DM [11,12,13]. There are studies showing that low-CHO diets have no significant effect on HbA1c as well as effectiveness [14]

In our study, no significant difference was observed in terms of GV; mean blood glucose levels; and TIR, TBR, and TAR values in the evaluation of the groups using CMGs. In the evaluation of the glucose AUC averages, no statistically significant difference was observed between the groups. Increased amounts of fat and protein do not affect GV much. Additionally, Pedersen et al. showed that blood glucose levels significantly decreased within 5 h following a low CHO breakfast; however, CHO restriction had no effect on postprandial hyperglycemia and GV within 24 h [15]. In a similar study examining the effects of 30% and 50% CHO diets on glycemic control, the 50% CHO diet was associated with higher postprandial glucose variability at small meals (afternoon snacks and second breakfast) and higher postprandial peaks at other meals (breakfast and dinner) [16].

Fluctuations in glucose levels trigger oxidative stress and increase microvascular complication occurrence. GV is a more significant kind of data in glycemic control than the HbA1c level. Good metabolic control aims to keep the blood glucose fluctuations of individuals with diabetes as close to healthy individuals as possible. In our study, no statistically significant difference in terms of GV following four different dietary interventions was observed (*p* = 0.59); however, GV was <36% after D1 and D2 and >36% after D3 and D4. Moreover, increased CHO contents increased GV.

No statistically significant difference was noted between the diet groups in the TIR (70–180 mg/dL) results. The mean TIR was >70% after low-CHO D1 and D2 and <70% after high-CHO D3 and D4.

The study by Brand-Miller et al. showed that low-GI diets decreased HbA1c levels in individuals with T1DM by 0.4% and in individuals with T2DM by 0.2% compared with high-GI diets [17]. Another study showed that the blood glucose level after a high-GL meal was higher than a meal with low GL [18]. GL affects individuals differently based on their BMI; the blood glucose level after a meal with a high GL was more prominent in the group with a high BMI.

Although a significant difference was noted between the intervention groups according to the GI and GL distributions of the meals in the current study, no significant difference was observed in the GV in the CGM results. Although the GI values of the diet menus were planned to be low in D1 and D3, and high in D2 and D4, all meals had a high GL.

Although the role of dietary fat and circulating FFAs in the pathogenesis of T2DM received considerable attention, relatively little attention has been paid to the possible consequences of FFA-induced insulin resistance in T1DM treatment [17]. Insulin resistance can also develop in individuals with T1DM. Overt hepatic insulin resistance in individuals with poorly controlled T1DM was suggested, and the reason for this is that the effect of insulin is suppressed by plasma FFAs [18,19]. Similarly, dietary fat and FFAs impair insulin sensitivity and increase hepatic glucose production [20,21]. Pharmacological interventions that reduce FFA levels in nondiabetic and type 2 diabetic individuals provide both improved insulin sensitivity and glucose tolerance [22]. In the present study, a statistically significant difference was observed between the dietary intervention groups regarding FFAs, whereas a statistically significant difference was noted only between the D1 and D4 groups in the pairwise comparison of the groups. D1 had a significantly higher mean FFA level than D4. The group with a high CHO intake and high GI had a low FFA level. The expected decrease in the amount of CHOs in the current study was compatible with the increase in FFA levels. In the literature, a diet with a high CHO intake and low GI can positively affect insulin sensitivity [23].

When we examined the effects of the diets on plasma TG levels, a statistically significant difference was observed between the groups. The amount of CHO and GI in the diet alone did not influence blood glucose in the short term; however, other parameters, including FFAs and TGs, were consistent with those in the literature with a low-CHO, low-GI diet model.

In our study, no statistically significant difference was observed in ketone levels after D1 and D2 interventions, indicating that diets containing 40% CHOs do not increase the risk of ketones. Therefore, consistent with the literature, we associated the lack of a significant decrease in the amount of CHOs with the absence of a significant difference in ketone levels. However, as ketone measurement was only taken in the urine, the ketone values that may have increased in the blood were considered a limitation of our study.

Fructosamine is a test method used in short-term follow-up and applications since it represents 2–3 weeks of glycemia before measurement. Fructosamine is formed by nonenzymatic glycation and is increased at high glucose concentrations. However, it is not included in evidence-based follow-up when monitoring glycemic control [24,25]. In the current study, we measured fructosamine levels to observe the effects of CHO administration on glycemic control in the short term since four different diet models were administered in the 4-week dietary intervention. In a 12-week intervention study conducted in patients with T2DM, the mean fructosamine levels in the low GI group at the early interim follow-up were significantly reduced compared with those of individuals using conventional CHO replacements [26]. In the current study, no statistically significant difference in terms of fructosamine levels was noted following dietary interventions compared with those at baseline. Contrary to the literature, the lack of significant results in our study once again draws attention to the fact that dietary interventions are short-lived and that they do not have a definite place in screening.

Although some studies showed that low-GI diets provide small but significant improvements in metabolic control as well as in body weight, BMI, low-density lipoprotein cholesterol (LDL-C) and total cholesterol, TG, and blood pressure levels [27,28], other studies asserted that low-GI diets reduced total cholesterol and LDL-C levels but had no effect on HDL or TG levels [29]. In our study, the lowest TG values were in diets with similar high GIs but different CHO contents (40% CHO and 60% CHO).

GV emerges as an independent risk factor for diabetes complications, with postprandial hyperglycemia being a significant contributor. Reducing CHO intake and following low-CHO diets are frequently advocated to prevent and manage diabetes. However, limiting or eliminating CHO may not be a long-term and sustainable approach for everyone. Alternatively, nutritional strategies for modulating glycemia may focus on the GI of foods and the GL of meals. Furthermore, studies showed that encouraging the consumption of meals with low GI and GL can be an essential support in reducing glucose levels and GV [30,31]. The fact that the results of our study were inconsistent with those from the literature was because we kept the number of cases limited in terms of economic cost and because of the difficulties that may occur in compliance with the diet. Again, it was believed that individuals with T1DM had the self-confidence to change their insulin doses on their own; however, it was underlined that no correction doses were used for regulating blood glucose, which possibly affected the results of our study.

In the current study, a statistically significant difference was noted in the increase in insulin doses when low-GI and high-GI diets were compared. Again, when the total (basal + bolus) insulin doses of low-GI and high-GI diets were compared, insulin doses were significantly higher in the high-GI diet. To improve metabolic control in individuals with T1DM, applying the insulin dose according to meal size and content is recommended. Carbohydrate counting has long been an essential treatment component in postprandial glucose control; however, the effects of carbohydrate counting and other dietary models on insulin dose increase should not be ignored.

Volunteers with T1DM over the age of 18 who applied to the endocrinology outpatient clinic were included in this study. However, the fact that the working duration was 4 consecutive weeks discouraged the participants regarding the volunteering principle. The wearing of sensors and provision of appropriate food service in this process facilitated the adaptation of the participants; however, the standardization of the menus was not well received by them. Although the participants consented to volunteer, the possibility of them not complying with the entire menu at home remains a limitation of the study.

Nutritional intervention, appropriate physical activity, and indirect weight control and medical treatment are the main steps of postprandial glycemic control. Low-GI diets are useful in controlling plasma glucose levels. In particular, a low-GI diet has positive effects on CVD risk, as well as controlling plasma glucose [32]. The presence of endogenous (insulin reserve and disease stage) and exogenous (drugs, diet, and physical activity) effects that affect GV, when considered holistically; the evaluation of all macronutrients individually; and increasing the number of cases may provide more comprehensive results. As reducing CHO intake increases FFA levels, monitoring the longer-term effects of this type of diet approach to prevent CVD risk is significant. A similar study suggested that CHO restriction and increased fat/protein intake can have long-term effects in terms of CVD [33].

## 5. Conclusions

In this study, FFA levels were observed in the group containing 40% CHO and having a low GI. Simultaneously, the D1 model was the group that contained the highest amount of cholesterol and saturated fat. High FFA levels in the blood can contribute to insulin resistance, which can be particularly challenging for individuals with T1DM. When the diet samples included in the study were examined, the D3 model was the richest meal in terms of soluble and insoluble fiber. It was the most suitable diet model for the Mediterranean nutritional model. Emphasizing nutritional recommendations in accordance with the Mediterranean nutritional model in the nutrition education of individuals with diabetes is significant.

## Figures and Tables

**Figure 1 nutrients-16-01383-f001:**
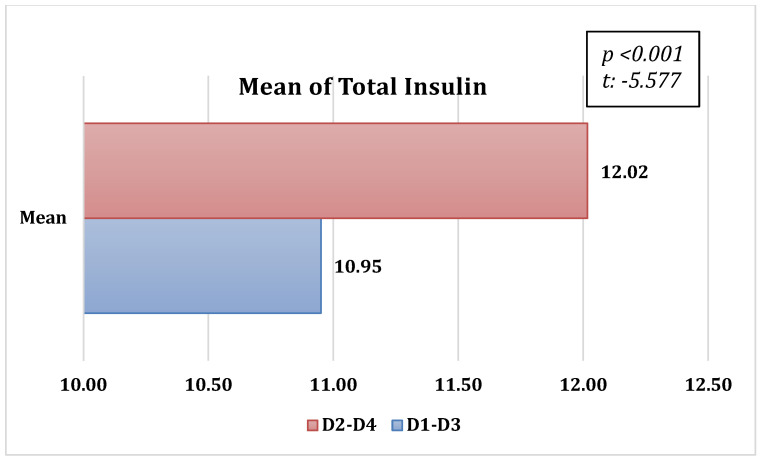
Insulin doses (Unite) administered in the diet groups D2-D4 and D1-D3 (t: paired samples *t*-test).

**Table 1 nutrients-16-01383-t001:** Sociodemographic characteristics of the participants (*n* = 17).

Variables	
Male, *n* (%)	9 (52.9)
Female, *n* (%)	8 (47.1)
Age (years)	29.7 ± 10.0
Diabetes duration (years)	11.8 ± 7.7
Hemoglobin A1c (%)	7.0 ± 0.9
Systolic blood pressure (mmHg)	114.7 ± 11.5
Diastolic blood pressure (mmHg)	72.4 ± 8.0
Smoking, *n* (%)	4 (23.5)
Alcohol use, *n* (%)	9 (52.9)
Body weight (kg)	72.5 ± 17.6
BMI (kg/m^2^)	24.2 ± 3.8
Body fat mass (kg)	15.5 ± 8.5
Body fat percentage (%)	21.2 ± 9.4
Lean body mass (kg)	57.0 ± 15.4

Data are presented as means ± SDs unless stated otherwise. BMI, body mass index.

**Table 2 nutrients-16-01383-t002:** Distribution of anthropometric measurements of the participants by gender.

Measured Parameter	Male		Female		*p*	Post Hoc Comparisons
	Mean ± SD	Min–Max	Mean ± SD	Min–Max		
Body weight at baseline (kg)	79.4 ± 15.1	54.5–96.0	62.7 ± 17.0	40.0–93.5	Group: F = 10.750, *p* = 0.003	D1–D2: *p* = 0.008
Body weight after D1 + D2 (kg)	79.4 ± 14.9	55.9–96.6	61.8 ± 16.6	39.5–92.0	Gender: F = 4.980, *p* = 0.042	D2–D3: *p* **< 0.001 ***
Body weight after D3 + D4 (kg)	78.6 ± 15.0	54.9–96.6	61.3 ± 16.5	39.4–91.7	Group*Gender: F = 2.170, *p* = 0.16	
BMI at baseline (kg/m^2^)	24.7 ± 3.4	17.6–28.7	23.4 ± 4.4	16.4–28.9	Group: F = 2.327, *p* = 0.12	
BMI after D1 + D2 (kg/m^2^)	24.7 ± 3.3	18.0–28.8	23.1 ± 4.2	16.2–28.4	Gender: F = 0.560, *p* = 0.47	
BMI after D3 + D4 (kg/m^2^)	24.5 ± 3.3	17.9–28.8	23.2 ± 4.6	16.2–28.9	Group*Gender: F = 1.426, *p* = 0.26	
Body fat percentage at baseline (%)	16.6 ± 6.2	7.7–24.1	27.7 ± 9.8	12.5–42.0	Group: F = 1.111, *p* = 0.34	
Body fat percentage after D1 + D2 (%)	16.8 ± 7.1	5.8–24.8	27.1 ± 9.8	12.6–42.7	Gender: F = 7.050, *p* = 0.018	
Body fat percentage after D3 + D4 (%)	16.5 ± 6.4	5.8–25.3	26.6 ± 10.0	11.0–42.5	Group*Gender: F = 0.777, *p* = 0.47	
Body fat mass at baseline (kg)	13.3 ± 5.8	5.3–20.7	18.7 ± 11.1	5.0–39.3	Group: F = 2.048, *p* = 0.15	
Body fat mass after D1 + D2 (kg)	13.6 ± 6.9	4.7–23.4	18.1 ± 11.1	5.0–39.3	Gender: F = 1.281, *p* = 0.28	
Body fat mass after D3 + D4 (kg)	13.2 ± 6.2	5.4–21.5	17.6 ± 11.1	4.3–39.0	Group*Gender: F = 1.770, *p* = 0.19	
Lean body mass at baseline (kg)	66.1 ± 13.0	44.1–87.7	44.0 ± 6.3	35.0–54.2	Group: F = 1.270, *p* = 0.30	
Lean body mass after D1 + D2 (kg)	65.8 ± 12.5	45.5–88.5	43.7 ± 6.2	34.5–52.7	Gender: F = 17.660, *p* = 0.001	
Lean body mass after D3 + D4 (kg)	65.5 ± 12.7	44.5–87.2	43.7 ± 6.2	35.1–52.7	Group*Gender: F = 0.179, *p* = 0.84	

* *p* < 0.001, F: repeated measurements ANOVA/significant *p* values are written in bold.

**Table 3 nutrients-16-01383-t003:** Biochemical and CGM values at baseline and after dietary interventions.

Variable	Baseline	Post-Diet 1	Post-Diet 2	Post-Diet 3	Post-Diet 4	*p*	Pairwise Comparisons
Fructosamine (μmol/L) mean ± SD (min–max)	0.4 ± 0.1 (0.3–0.4)	NA	0.3 ± 0.0 (0.3–0.4)	NA	0.3 ± 0.0 (0.3–0.4)	F = 2.622, *p* = 0.108	
Triglycerides (mg/dL) mean ± SD (min–max)	73.6 ± 31.3 (29–143)	134.3 ± 91.9 (32–334)	109.4 ± 65.7 (39–282)	118.8 ± 53.5 (38–243)	97.5 ± 61.0 (36–278)	F = 6.696, *p* **= 0.002 ***	D1-D2: *p* = **0.049 *** D1-D4: *p* = **0.006 ***
Presence of ketone bodies (%)	29.41	5.88	11.77	NA	NA	z = 0.600, *p* = 0.545	
Ketone bodies (mg/dL) mean ± SD (min–max)	15.88 ± 38.29 (0–150)	0.88 ± 3.64 (0–15)	1.76 ± 4.98 (0–15)	NA	NA	t = 0.588, *p* = 0.561	
Free fatty acids (nmol/mg) mean ± SD (min–max)	1.19 ± 0.21 (0.90–1.70)	1.60 ± 0.57 (0.80–2.99)	1.36 ± 0.53 (0.88–2.16)	1.41 ± 0.42 (0.83–2.16)	1.23 ± 0.53 (0.72–2.95)	X^2^ = 11.88, *p* = 0.018	D1-D4: *p* = **0.014 ***
Average blood glucose (mg/dL) mean ± SD (min–max)	NA	135.2 ± 23.0 (88–168)	138.2±31.1 (88–189)	141.4±26.5 (101–192)	146.8 ± 30.3 (85–213)	F = 1.329, *p* = 0.276	
GV (%)	NA	34.2 ± 9.8 (14.9–52.7)	34.4 ± 8.8 (13.6–49.6)	36.3 ± 9.5 (20.0–57.4)	36.8 ± 8.2 (15.3–46.7)	F = 0.646, *p* = 0.589	
TIR (%), mean ± SD (min–max)	NA	71.7 ± 14.2 (47.7–100.0)	70.2 ± 16.8 (42.3–97.9)	69.8 ± 15.4 (39.4–95.8)	66.8 ± 14.5 (38.1–95.6)	F = 0.683, *p* = 0.567	
TBR (%), mean ± SD, median (min–max)	NA	5.8 ± 6.5 3.75 (0–22.7)	4.7 ± 3.7 4.3 (0–14.56)	5.7 ± 3.8 5.46 (0–12.02)	5.3 ± 3.8 4.15 (0–11.45)	X^2^ = 0.479, *p* = 0.923	
TAR (%), mean ± SD, median (min–max)	NA	10.2 ± 6.5 10.2 (0–20)	11.2 ± 9.2 7.4 (0–28.8)	11.9 ± 8.0 10.8 (0–27.2)	13.5 ± 7.9 11.9 (0–28.7)	F = 1.172, *p* = 0.330	
Blood glucose AUC (mg/dL), mean ± SD, median (min–max)	NA	83,236.56 ± 305 9.84 (24,176–135,046)	86,380.94 ± 377 9.28 (14,298–147,814)	88,088.06 ± 3091 0.42 (45,182–142,304)	89,496.81 ± 323 5.43 (13,187–152,728)	F = 0.359, *p* = 0.783	

AUC, area under the curve; GV, glycemic variability; NA, not available; SD, standard deviation; TAR, time above range; TBR, time below range; TIR, time in range. F, repeated measurements ANOVA; X^2^, Friedman’s test; * significant *p* values are written in bold. Post-diet data are expressed as means ± SDs unless stated otherwise.

**Table 4 nutrients-16-01383-t004:** Comparison of dietary intervention measurements.

	Mean ± SD	Median (Min–Max)	Test Statistics
D1D2_AG	136.68 ± 25.71	141.0 (91.5–172.0)	t = −1.476 *p* = 0.16
D3D4_AG	144.12 ± 25.19	145.0 (93.0–198.5)	
D1D3_AG	138.29 ± 22.62	138.5 (98.0–175.5)	t = −1.018 *p* = 0.32
D2D4_AG	142.50 ± 26.70	141.5 (86.5–189.5)	
D1D2_GV	34.27 ± 8.00	36.67 (14.84–44.80)	t = −1.235 *p* = 0.24
D3D4_GV	36.52 ± 7.78	38.32 (17.66–49.73)	
D1D3_GV	35.22 ± 8.11	35.31 (18.07–48.84)	W = −0.024 *p* = 0.98
D2D4_GV	35.56 ± 7.31	37.63 (14.44–43.36)	
D1D2_TIR	70.97 ± 13.29	74.00 (50.82–98.06)	t = 1.409 *p* = 0.18
D3D4_TIR	68.3 ± 12.56	67.96 (41.17–95.72)	
D1D3_TIR	70.78 ± 13.00	73.39 (49.47–97.03)	t = 1.138 *p* = 0.27
D2D4_TIR	68.48 ± 13.02	70.64 (47.35–96.76)	
D1D2_TBR	4.27 ± 3.23	3.11 (0.00–11.38)	t = 0.751 *p* = 0.46
D3D4_TBR	3.66 ± 1.75	3.81 (0.49–6.16)	
D1D3_TBR	4.01 ± 2.76	2.66 (0.94–8.83)	W = −0.024 *p* = 0.98
D2D4_TBR	3.92 ± 2.30	3.87 (0.46–9.21)	
D1D2_TAR	10.67 ± 7.16	10.08 (0.00–21.76)	t = −1.700 *p* = 0.11
D3D4_TAR	12.72 ± 6.63	12.33 (0.00–25.74)	
D1D3_TAR	11.03 ± 6.71	10.49 (0.00–21.28)	t = −1.066 *p* = 0.30
D2D4_TAR	12.36 ± 7.16	10.42 (0.00–23.94)	
D1D2_TG	121.82 ± 75.76	94.5 (35.5–292.5)	W = −1.279 *p* = 0.20
D3D4_TG	108.12 ± 53.51	99.0 (37.0–260.5)	
D1D3_TG	126.53 ± 70.90	110.0 (35.0–273.0)	W = −2.296 *p* **= 0.022**
D2D4_TG	103.41 ± 61.44	78.0 (37.5–280.0)	
D1D2_FFA	1.48 ± 0.51	1.43 (0.86–2.90)	W = −1.894 *p* = 0.06
D3D24_FFA	1.32 ± 0.40	1.27 (0.78–2.56)	
D1D3_FFA	1.51 ± 0.45	1.54 (0.82–2.58)	W = −1.870 *p* = 0.06
D2D4_FFA	1.29 ± 0.49	1.13 (0.82–2.88)	

AG, average blood glucose; D1, diet 1; D2, diet 2; D3, diet 3; D4, diet 4; FFA, free fatty acid; GV, glycemic variability; SD, standard deviation; TAR, time above range; TBR, time below range; TG, triglyceride; TIR, time in range; t, paired samples *t*-test. Significant *p* values are written in bold.

**Table 5 nutrients-16-01383-t005:** Insulin doses administered during different meals.

	Morning Preprandial Insulin Dose (IU) $\bar{X}\pm SD$ (Min–Max)	Noon Preprandial Insulin Dose (IU) $\bar{X}\pm SD$ (Min–Max)	Evening Preprandial Insulin Dose (IU) $\bar{X}\pm SD$ (Min–Max)	Basal Insulin Dose (IU) $\bar{X}\pm SD$ (Min–Max)
Diets with low GI (D1 and D3)	8.1 ± 5.4 (2–19)	8.4 ± 4.4 (4–18)	8.9 ± 4.0 (5–18)	18.1 ± 6.2 (10–33)
Diets with high GI (D2 and D4)	9.6 ± 5.4 (4–20)	9.9 ± 4.5 (5–20)	10.3 ± 4.4 (6–21)	18.6 ± 6.6 (10–35)
Test statistics, *p* value	t = −5.258, *p* < 0.001 *	t = −4.557, *p* < 0.001 *	t = −3.748, *p* = 0.002 *	t = −2.256, *p* = 0.041

GI, glycemic index; SD, standard deviation. * *p* < 0.001, t, paired samples *t*-test. Significant *p* values are written in bold.

## Data Availability

The data obtained in this study are available from the corresponding author upon request. The data are not publicly available due to subjects’ privacy.

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
