# Peer review of "Effects of Dietary Carbohydrate Concentration and Glycemic Index on Blood Glucose Variability and Free Fatty Acids in Individuals with Type 1 Diabetes"

_nutrients, 2024, doi:10.3390/nu16091383_

Round 1

Reviewer 1 Report

Comments and Suggestions for Authors

Comments:

1.      The concept is not novel. There are plenty of articles regarding this concept published already, and some of the articles are large scale meta-analysis (eg: Nutrients. 2022 Jan; 14(2): 307.) or even umbrella reviews (Nutrients 2023, 15, 861.). This study is just a single study with 17 participants making it neither representative nor innovative. Please describe the strength of current study.

2.      Current study design is to observe the short-term effects of four different diet models on blood glucose, glycemic variability and FFA levels with 5 days of administration and 2 days of wash-out. This study design is hard to resolve the question of the manuscript title as “ LOW GLYCEMIC INDEX OR LOW CARBOHYDRATE: WHICH ONE MATTERS?”. The sort-term four different diet models design is not easy to provide scientific information to this question. The conclusion as “Different diets may increase the risk of cardiovascular disease by affecting GI, FFA levels, and blood glucose.” could subjective be accepted. Thus, title may be revised in a more preserved form according to the results only.

3.          The article is entitled “low carbohydrate”, but the CHO percentage of total calory intake of this study is 40%, which does not meet the low-carb definition. According to the definition, this is a moderate-carbohydrate diet. Please confirm the definition used in current study.

4.      Many typing error should be thoroughly re-check.

A.      One example is “Insulin” is not spelled correctly in Figure 1, and this figure list in current position is not suitable. Figure 1 is also not easy to catch up and may be misleading as insulin dosage administered before meal.

B.       

Line 35 (Abstract) mentioned about “total glucose”à What is total glucose??

Line 134 (Material) The GI in all diet groups was significantly different (p <0.001). The afternoon snack had a significance of p=0.022. à Should this belong to the results section? And why is the GI different in all diet groups?

Line 152 (Results) At the beginning of the study, the patients had a diabetes duration of 11.8 ± 7.7 years à I believe DM duration would not change much at the end of the study.

Line 168 the baseline body fat mass for females 44.0 ±6.3 kg?

Line 231-232 Table 1, Fat-free body mass? = Lean body mass?

5.      The conclusion section is too long and with no highlights. Please downsize and condense this section.

6.      The citation articles listed in this manuscript were not new, and many concepts or information may not be suitable.

Comments on the Quality of English Language

Many typing error should be thoroughly re-check.

Author Response

Dear Editors and  Reviewers,

First of all, we would like to thanks you for your valuable opinions on the development of article,your efforts in this processand your seminal comments. We have tried to respond to your feedback in the best way possibble and we have itemized your suggestions ın the uploaded file. 

If we have any shortcomings in our work, we will gladly reply again.

Sincerely yours.

Reviewer 2 Report

Comments and Suggestions for Authors

Comments:

The title accurately reflects the paper's subject, and the abstract is concise and contains enough information to highlight the article's content. The introduction briefly reviews the existing literature but does not clarify the problem studied in this article. This section should be improved using more references, preferably as recent as possible. The information provided by the authors regarding the working methodology is enough to allow the repetition of the experiment. However, the abstract mentions that the subjects consumed the diets for four days and in Materials and Methods for four weeks. The authors should clarify this.

Moreover, in the Discussions chapter, some results are presented in text and tables. The authors should present them in a different form than they currently do. Also, the conclusions are not presented clearly in the Conclusions chapter. The experiment's result does not appear succinct.

Author Response

Dear Editors and Reviewers, First of all, we would like to thank you for your valuable opinions on the development of article, your efforts in this process and your seminal comments. We have tried to respond to your feedback in the best way possible and we have itemized your suggestions in the uploaded file. 

If we have any shortcomings ın our work, we will gladly reply again.

Sincerely yours

Round 2

Reviewer 1 Report

Comments and Suggestions for Authors

None

Author Response

Dear editors, in the last check, there was a reference, but it seemed to be missing in the article, we rearranged it,we uploaded it again, it is important for you to consider, thank oyu very much for your interest and support. 

Best regards 
